# 3D Image Reconstruction
# from Multi-focus Microscopic Images [*]

Takahiro Yamaguchi[1], Hajime Nagahara[2], Ken'ichi Morooka[1],
Yuta Nakashima[2], Yuki Uranishi[2], Shoko Miyauchi[1], and Ryo Kurazume[1]

[1] Graduate School of Information Science and Electrical Engineering
Kyushu University, Fukuoka 819-0395, Japan `morooka@ait.kyushu-u.ac.jp`
[2] Institute for Datability Science, Osaka University, Osaka 565-0871, Japan
`nagahara@ids.osaka-u.ac.jp`

**Abstract.** This paper presents a method for reconstructing 3D image
from multi-focus microscopic images captured with different focuses. We
model the multi-focus imaging by a microscopy and produce the 3D im-
age of a target object based on the model. The 3D image reconstruction
is done by minimizing the difference between the observed images and
the simulated images generated by the imaging model. Simulation and
experimental result shows that the proposed method can generate the
3D image of a transparent object efficiently and reliably.

**Keywords:** 3D imaging · Microscopy · Multi-focus images · Transparent
object.

## 1    Introduction

Cell observation by optical microscopy is widely used in biology, medicine, and
so on. For example, cytodiagnosis and iPS cells culture are based on the cell
observation. A regular microscopy acquires a 2D image of a target cell with
a 3D structure. However, only a part of slices of the 3D cell structure can be
observed as a focused image because the depth of field of the general microscope
is narrow. Under such circumstances, various applications can be expected for
the measurement technology of 3D cell structure.

A simple way to get the 3D structure using a microscopy is to stack multiple
slice images with different focuses. Various methodologies[1] have been proposed
to measure the multi-focus images. However, the simple stacked multi-focus im-
ages include many unclear regions because of reflections of front and backside
of the focus. To solve this problem, a confocal microscope is often used. In a
confocal microscope, a pinhole in front of a light detector cuts off light that is
out of focus while allowing only the fluorescence light from the in-focus spot to
enter the light detector. Thus, a confocal microscope is a useful and powerful
tool to obtain clear images of only in-focus regions.

[*] Supported by JST CREST Grant Number JPMJCR1786 and JSPS KAKENHI
Grant Number JP19H04139, Japan.

Another approach for imaging the 3D object structure is Computed tomography (CT)[2]. In CT scanning, when X-rays is irradiated to a target object, X-ray CT detects X-ray passed through the object by a detector located on opposite sides of the object. Here, it is assumed that X-ray is absorbed and attenuated by the object in the irradiation. On the assumption, many intensities of X-ray are measured by rotating a pair of X-ray source and detector around the object. When a target object is represented with a set of voxels, the measured intensities are used to estimate the attenuation coefficient of each voxel.

Similar to X-ray CT, Optical Projection Tomography (OPT)[3] have been proposed for microscopic 3D imaging. Using regular light, lens optics and silicon image sensor, OPT estimates the light attenuation of each voxel. However, since the imaging systems of X-ray and OPT requires rotation mechanisms, the methodology of X-ray CT and OPT is directly inapplicable to regular microscope with no rotation mechanism.

In this paper, we propose a method of 3D image from multi-focus microscopic images obtained with different focuses. We model an imaging system for acquiring the multi-focus microscopic images with different focuses. In the imaging system, the microscopic images are produced by the light emitted from its light source. When the light passes through the transparent object, the light is attenuated depending on the transmittance of the object material. This means that each pixel in the microscopic image is related to these attenuated light. Considering this, we reconstruct the 3D image of the object by minimizing the difference between the observed images and the simulated images generated by the imaging model.

Similar with our method, there are two approaches for 3D imaging using multi-focus images. The first is the reconstruction of 3D image which contains appearance of inner slices of transparent objects[4]. The 3D image is generated by simply piling these discrete slices acquired by CCD cameras. The second approach is to reconstruct 3D image of a target object's luminescence from multi-focus images obtained by a fluorescence microscope[5]. Unlike the two approaches, we aim to reconstruct the 3D image as a set of voxels having transmittance values from multi-focus images obtained by a general bright field microscopy.

## 2   3D image reconstruction from multi-focus images

### 2.1   Imageing model for multi-focus microscopic images

Fig. 1 shows how the pixel intensity is observed by a microscopy. We denote an intensity value of an arbitrary pixel $(x, y)$ in the $s$-th image in the sequence of the multi-focus microscopic images as $I_{\boldsymbol{\alpha}}(x, y, s)$. We assume that a target space including one or more than cells is represented by a set of voxels. 3D image is estimated as transmittances of the voxels $\alpha_i$ $(i = 0, 1, \cdots, N_v - 1)$ as shown in Fig. 1. An incident light is emitted from a light source under a stage.

The incident light is discretized as a set of $N_r$ discrete rays. The intensity $l_j$ of the $j$-th $(j = 0, 1, \cdots, N_r - 1)$ ray is attenuated every time the ray passes

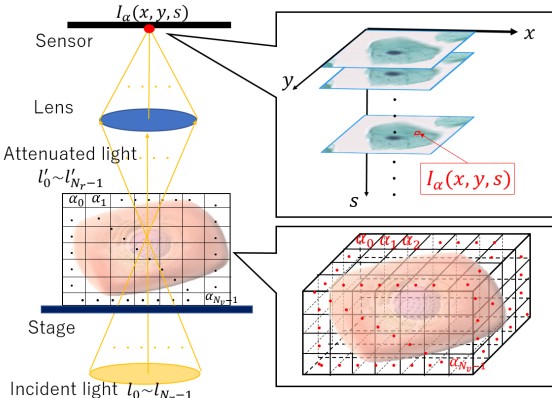

**Fig. 1.** Imaging system model

through each voxel. The attenuation is affected by the transmittance of the voxel and the length of the ray through the voxel. Hence, we model the relationship between $l_j$ and the attenuated ray $l'_j$ by

$$l'_j = l_j \times \prod_i \alpha_i^{d_{ji}}, \tag{1}$$

where $d_{ji}$ is the length of the $j$-th ray if the ray passes through the $i$-th voxel. Otherwise, $d_{ji} = 0$.

By taking the log of both side in Eq. (1) and expanding the log, Eq. (1) is rewritten as

$$\log l'_j = \log l_j + \sum_i d_{ji} \log \alpha_i. \tag{2}$$

By collecting all relationships between the incident ray and attenuated ray, $l_j$ and $l'_j$ using Eq. (2), we obtain the following formulation:

$$\boldsymbol{L'} = \boldsymbol{DA} + \boldsymbol{L}, \tag{3}$$

where

$$\boldsymbol{L} = \begin{bmatrix} \log l_0 \\ \vdots \\ \log l_{N_r-1} \end{bmatrix}, \ \boldsymbol{L'} = \begin{bmatrix} \log l'_0 \\ \vdots \\ \log l'_{N_r-1} \end{bmatrix}, \ \boldsymbol{A} = \begin{bmatrix} \log \alpha_0 \\ \vdots \\ \log \alpha_{N_v-1} \end{bmatrix},$$

$$\boldsymbol{D} = \begin{bmatrix} d_{00} & \cdots & d_{0(N_v-1)} \\ \vdots & \ddots & \vdots \\ d_{(N_r-1)0} & \cdots & d_{(N_r-1)(N_v-1)} \end{bmatrix}.$$

Here, we assume that aperture of the light source and objective lens are enough large to the target cell. On this assumption, an arbitrary pixel $I_{\boldsymbol{\alpha}}(x, y, s)$ can be similarly expressed by shifting the stage along with $(x, y, s)$ coordinates. On the

assumption, $\boldsymbol{D}$ and $\boldsymbol{L}$ are regarded as the function of the three parameters $x$, $y$, and $s$. We modify $\boldsymbol{D}$ is the function of $(x, y, s)$ and the components of matrix D is shifted by $(x, y, s)$. Hence, Eq. (3) is rewritten by

$$\boldsymbol{L}'(x, y, s) = \boldsymbol{D}(x, y, s)\boldsymbol{A} + \boldsymbol{L}. \tag{4}$$

Finally, $I_{\boldsymbol{\alpha}}(x, y, s)$ is calculated by the total amount of the attenuated rays $l'_j$:

$$I_{\boldsymbol{\alpha}}(x, y, s) = \sum_j l'_j(x, y, s). \tag{5}$$

## 2.2   Estimation of the voxel transmittance

Using the model as mentioned in Sec 2.1., we simulate the observed multi-focus images. When the estimated transmittances of the target voxels are close to the real ones, the intensity value of the simulated multi-focus images $I_{\boldsymbol{\alpha}}(x, y, s)$ should be the same as the intensity value of the observed images $I(x, y, s)$ by microscopy. Considering this, the 3D image is reconstructed by minimizing an objective function $F$:

$$F(\boldsymbol{\alpha}) = E(\boldsymbol{\alpha}) + wTV(\boldsymbol{\alpha}), \tag{6}$$

where $\boldsymbol{\alpha}$ is a vector composed of all the transmittances $\boldsymbol{\alpha} = (\alpha_0, \alpha_1, \cdots, \alpha_{N_v-1})$. The parameter $w$ is a weighted coefficient as a regularization parameter. The conjugate gradient method is applied to find optimal transmittances which minimize $F(\boldsymbol{\alpha})$. The function $E(\boldsymbol{\alpha})$ in Eq. (6) represents the difference between the intensity value $I(x, y, s)$ in the observed image and $I_{\boldsymbol{\alpha}}(x, y, s)$ in the simulated image by Eq. (5). The function $E$ is defined as

$$E(\boldsymbol{\alpha}) = \sum_s \sum_x \sum_y (I_{\boldsymbol{\alpha}}(x, y, s) - I(x, y, s))^2. \tag{7}$$

On the contrary, $TV(\boldsymbol{\alpha})$ is a regularization function base on a total variation (TV) norm to reconstruct the 3D image smoothly. Practically, the value of $TV(\boldsymbol{\alpha})$ is calculated by the total transmittance difference between the traget voxels and its six neighbor voxels:

$$TV(\boldsymbol{\alpha}) = \sum_{k \in \Phi_i} (\alpha_i - \alpha_k)^2, \tag{8}$$

where $\Phi_i$ is the set of the six neighbors of the target $i$-th voxel.

## 2.3   Efficient search of optimal transmittances

From Eq. (6), our proposed method finds the optimum transmittances by iteratively updating the transmittances. To find the optimum efficiently and robustly, we introduce the two followings.

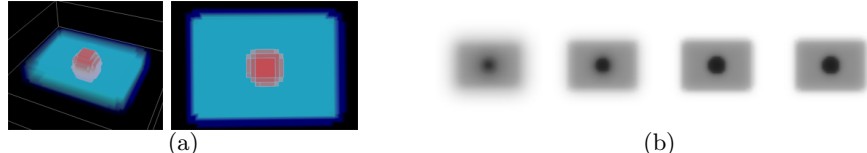

(a)                                                    (b)

**Fig. 2.** (a)An artificial 3D cell model; (b) multi-focus images of the cell model.

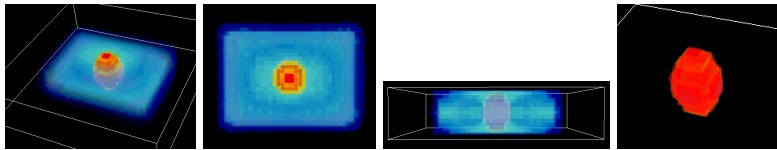

**Fig. 3.** 3D image of the cell model estimated by our method.

**Initialization from input images:**  The initial values of the transmittances are important to find the optimum transmittances robustly by the conjugate gradient method. Given a sequence of $N_s$ multi-focus images, we determine the initial transmittances based on the intensity value $I(x, y, s)$ of the original image sequence.

Let us consider that all rays are intersected at the $i$-th voxel when the intensity $I(x, y, s)$ is calculated. In this case, since all rays pass through the $i$-th voxel, the transmittance $\alpha_i$ of the $i$-th voxel strongly influences on the calculation of $I(x, y, s)$ compared with other voxels. Moreover, in our imaging system model, each ray passes through at least $N_s$ voxels. Therefore, the optimal transmittance value of the $i$-th voxel is approximately regarded as the $N_s$-th root of $I(x, y, s)$. Considering these, the initial transmittance value $\alpha_i^{(0)}$ of the $i$-th voxel is calculated by

$$\alpha_i^{(0)} = \sqrt[N_s]{I(x, y, s)}. \tag{9}$$

**Coarse-to-fine search:**  From Eq.(1) - (5), the computational burden in our method depends on the number of rays. When the small number of the rays is used, the estimation of the transmittances can be speeded up. However, the light is discretized roughly by the small number of the rays. Therefore, the use of such rays results in the low accuracy of estimating the transmittances. On the other hand, in the case of using many rays, although the estimation of the transmittances is time-consuming, the reliable transmittances can be obtained.

Considering the trade-off between the efficiency and accuracy of estimating the transmittances, we introduce a coarse-to-fine strategy. Firstly, in the coarse step, the transmittances are roughly estimated by using a small number of the rays (in our case, $N_r = 25$). The obtained transmittances in the coarse step are used as the initial values of the transmittances in the following fine step. In the fine step, we find the optimal values of the transmittances by using many rays (in our case, $N_r = 533$).

## 3    Experimental results

To evaluate the performance of the proposed method, we made a simulation using synthetic cell images and experiments using real cell images. In the simulation and the experiment, the parameter $w$ in Eq.(6) is set to $w = 0.125$.

### 3.1    Simulation using synthetic cell images

In the simulation, we generate two virtual 3D cell models. Fig. 2(a) shows one of them. It consists of a nucleus (red in Fig. 2(a)), a cytoplasm(light blue) and a cell membrane (blue). From real cell images, it is observed that the transmittance values of the nucleus tend to be lower than those of the cyptoplasm and the membrane. Based on the observation, the transmittance values of the three components are set to 0.80 (the nucleus), 0.95 (the cyptoplasm), and 0.98 (the membrane), respectively.

The imaging system model (Section 2.1) is applied to generate the synthetic images of the virtual cells. Here, the number $N_r$ of the rays used in the 3D image reconstruction is set to 533 so that for each voxel shown in Fig. 1, at least one ray passes through the voxel when the maximum blur is occurred in the model. Finally, we obtain 11 multi-focus images with $50 \times 50$ [pixel] (Fig. 2(b)).

We verify initialization from input images and coarse-to-fine search(section 2.3). In the verification, the 3D images are reconstructed by the proposed methods. Moreover, the proposed methods are compared by the two methods. First one is the method in which all initial values of the transmittances are set to 0.8 through some preliminary experiments. Second one is the method which uses 25 or 533 rays to reconstruct the 3D image.

To measure the accuracy of the reconstructed 3D image, we use the root mean square error (RMSE) between the reconstructed 3D image and their ground truth values. Table 1 shows the average of RMSE and computational time between two virtual 3D cell models for the each method.

In the verification about the initialization, from Table 1, the initialization from input images improves the accuracy of reconstructing 3D image compared with the method in which all initial values of transmittances are set to 0.8. Moreover, the computational time for the initialization from input images is shorter than that of the method in which all initial values of transmittances are set to 0.8 using same number of the rays. From these results, the initialization from input images is useful for obtaining the reliable transmittances.

In the verification about the coarse-to-fine search, from Table 1, the proposed method using the coarse-to-fine search improves the accuracy of the reconstructing 3D image compared with the methods using only 25 or 533 rays. Moreover, the computational time of the methods using the coarse-to-fine search is shorter than the methods using only 533 rays.

The computational time in the 3D image reconstruction increases according to the number of the used rays in the reconstruction. In the coarse-to-fine search, the first coarse step is to search the optimal transmittances roughly by using a small number of the rays. In the second fine step, we find the optimal values of the

**Table 1.** Ablation study for initialization and coarse-to-fine methods.

| Initial values | $N_r$ | RMSE [$\times 10^{-2}$] | Time[sec] |
|---|---|---|---|
| Initialization from input images | coarse-to-fine (25 to 533) | 0.869 | 336 |
| Initialization from input images | 25 | 0.875 | 58 |
| Initialization from input images | 533 | 0.873 | 1,553 |
| Constant value ($\alpha_i^{(0)}$=0.80) | coarse-to-fine (25 to 533) | 0.878 | 373 |
| Constant value ($\alpha_i^{(0)}$=0.80) | 25 | 0.882 | 76 |
| Constant value ($\alpha_i^{(0)}$=0.80) | 533 | 0.906 | 2,209 |

transmittances by using many rays. Therefore, the coarse-to-fine search reduces the total number of the used rays in the reconstruction. Moreover, the coarse search enables to find the values of the transmittances closed to the optimal ones while avoiding local minimum. Owing to these, the proposed method using the coarse-to-fine search can find the optimal transmittances efficiently and stably.

Thus, the proposed method using the initialization from input images and the coarse-to-fine search achieves the best accuracy of the 3D images reconstruction while reducing the computational time drastically compared with the methods using only 533 rays. Fig. 3 shows the 3D image of the virtual cell estimated by the proposed method using the initialization from input images and the coarse-to-fine search.

## 3.2 Experiment using real cell images

In the experiment, the proposed method is applied to the multi-focus images of real cell images to reconstruct the 3D image of the real cells. Fig. 4 (a) and (b) show the multi-focus image sequences of normal and cancer cells. The size and spatial resolution of each cell image is $62 \times 62$ [pixel] and $0.92\mu m/1$pixel. To apply the proposed method, the color cell images are converted into the gray scale images.

Fig. 5 (a) and (b) show the 3D image of the normal and cancer cells reconstructed from Fig. 4, respectively. It takes about 1,650 [sec] on average to reconstruct 3D image. The average computational time in the experiments is longer than that in the simulation because of the following reason. In the simulation, we assume that the cytoplasm is homogeneous with no other components. In other words, all the voxels in the artificial cell model have almost the same transmittance values. On the contrary, a real cell contain other components such as mitochondria. This means that there are the voxels with various transmittance values in the real cells. Owing to the complex structure of the cell, reconstructing the 3D image of the real cells is time-consuming. One of our future works is to speed up the estimation of the transmittances of real cells with complex structures.

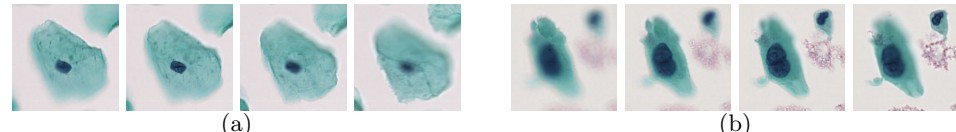

(a)                  (b)

**Fig. 4.** Multi-focus images of real cells: (a) a normal cell; (b) a cancer cell.

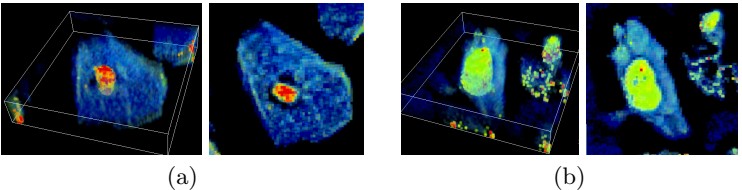

(a)                  (b)

**Fig. 5.** Reconstructed 3D images of (a)the normal and (b)cancer cells.

## 4   Conclusion

We proposed a method for reconstructing the 3D image of a transparent object from multi-focus microscopic images. To achieve this, we model a microscopic imaging system for acquiring the multi-focus microscopic images with different focuses. The optimal values of the transmittances are determined by minimizing the difference between the intensities of the observed image and the simulated image by our model. From the simulation using the virtual cell, it is confirmed that the proposed method can reconstruct the optimal 3D image efficiently and stably. In addition, the 3D image reconstruction from the real cell images is achieved with these proposed methods.

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
