# OpenReview forum: "3D Image Reconstruction from Multi-focus Microscopic Images"
_MICCAI.org/2019/Workshop/COMPAY — Submitted to COMPAY 2019_

### Official Review · AnonReviewer2 · 2019-07-29
**Interesting idea needing additional discussion**

**Rating:** 5
**Confidence:** 4

**Review:**

The authors present a technique for reconstructing 3D image from multi-focus microscopic images.
The approach is interesting, but the experimental design and manuscript require additional effort

The authors should discuss more fully the motivation for this work, why is it important? It is not sufficient to simply say “Under such circumstances, various applications can be expected for the measurement technology of 3D cell structure”, the burden is on the authors to put their work in the context of the field

What are some uses cases where this work can be employed? Further, it would make sense to pick one of these use cases and use it as a validation to get a feeling for if this approach “works”, i.e., is fit for purpose.

The discussion is regarding course-to-fine (3.1) is too long and doesn’t provide additional information over what is already discussed in body of the manuscript below equation 9. Given that course-to-fine methods are well used, the entire concept need not be introduced. Simply saying that finer takes longer computation time but results in a better estimate is sufficient
The discussion of x-ray/ CT is likely out of scope and the space could be better used otherwise (see below)

On the other hand, the authors use RMSE error for their approach. It is not obvious to me what a “good” versus bad RMSE error would be? Quite frankly, in this case I don’t know what the magnitude of such an error would be appropriate to differentiate an excellent approach from a poor approach. Typically one either compares their algorithm to another state of the art algorithm, and show superiority in that manner, or uses an intuitive measure (e.g., accuracy) in the context of a well known problem. Since this is not a well-known problem, I don’t think there is an intuitive measure to be found, and would thus suggest either comparing against another approach, or further showing comparative examples of RMSE error to give the reader a feeling to say “yes this approach is doing a good job”

In 3.2, I would expect to have more of a discussion about the quality of the results, with an attempt at convincing the reader that the results are sufficient for purpose. This discussion section focuses solely on the time component which is secondary to the actual performance of the approach.

Although the authors present synthetic data, this data seems to be too easy for the use case? They should make an effort to convince the reader that this synthetic data is appropriately challenging for this task, especially given that they use it to report metrics. Although the synthetic data is used for training the model, since there is no option for qualitative results on real-data, a I suspect a deeper investigation using varying synthetic data would be required.

Given that this appears to be a preliminary evidence type manuscript, I would expect to see significantly more discussion on potential limitations and challenges associated with this work. Does it only work for specific stains? Does it work on all organs? What if the slide comes from another lab with a slightly different thickness of tissue, does the algorithm need to be tuned? How?

Beyond qualitatively looking at the synthetic data, how do the authors know that these results are reasonable and appropriate?

---

### Official Review · AnonReviewer4 · 2019-08-15

**Rating:** 5
**Confidence:** 3

**Review:**

This paper presents a method to reconstruct 3D images from multi-focus microscopic images captured with different focuses. Results are presented on synthetic images as well as real images.

Comments:

•	Figure 1 is quite important and the caption should explain more details about it, it should be self-explanatory.

•	Several values are provided without a clear explanation, for example “value of transmittance set to 0.8 through some preliminary experiments”, or values of 25 and 533 rays. These values should be justified in the paper.

•	The proposed method and its mathematical formulation are interesting, but the paper lacks a practical application of this approach, as well as a comparison with other models, if available. The authors state in their conclusion that the proposed method can reconstruct the optimal 3D image efficiently and stably, but it is not clear what is the direct implication of this achievement in cancer research.